# PeerJ

# Disease dynamics and potential mitigation among restored and wild staghorn coral, *Acropora cervicornis*

Margaret W. Miller[1], Kathryn E. Lohr[2], Caitlin M. Cameron[1,2], Dana E. Williams[1,2] and Esther C. Peters[3]

[1] NOAA-National Marine Fisheries Service, Southeast Fisheries Science Center, Miami, FL, United States
[2] Rosenstiel School of Marine and Atmospheric Science, University of Miami, Miami, FL, United States
[3] Department of Environmental Science and Policy, George Mason University, Fairfax, VA, United States

## ABSTRACT

The threatened status (both ecologically and legally) of Caribbean staghorn coral, *Acropora cervicornis*, has prompted rapidly expanding efforts in culture and restocking, although tissue loss diseases continue to affect populations. In this study, disease surveillance and histopathological characterization were used to compare disease dynamics and conditions in both restored and extant wild populations. Disease had devastating effects on both wild and restored populations, but dynamics were highly variable and appeared to be site-specific with no significant differences in disease prevalence between wild versus restored sites. A subset of 20 haphazardly selected colonies at each site observed over a four-month period revealed widely varying disease incidence, although not between restored and wild sites, and a case fatality rate of 8%. A tropical storm was the only discernable environmental trigger associated with a consistent spike in incidence across all sites. Lastly, two field mitigation techniques, (1) excision of apparently healthy branch tips from a diseased colony, and (2) placement of a band of epoxy fully enclosing the diseased margin, gave equivocal results with no significant benefit detected for either treatment compared to controls. Tissue condition of associated samples was fair to very poor; unsuccessful mitigation treatment samples had severe degeneration of mesenterial filament cnidoglandular bands. Polyp mucocytes in all samples were infected with suspect rickettsia-like organisms; however, no bacterial aggregates were found. No histological differences were found between disease lesions with gross signs fitting literature descriptions of white-band disease (WBD) and rapid tissue loss (RTL). Overall, our results do not support differing disease quality, quantity, dynamics, nor health management strategies between restored and wild colonies of *A. cervicornis* in the Florida Keys.

## INTRODUCTION

Disease, in conjunction with co-occurring and likely interacting stressors such as storms and warming temperatures, is the major driving factor placing the Caribbean staghorn coral, *Acropora cervicornis*, at risk of extinction (reviewed in *Aronson & Precht, 2001*; *IUCN, 2012*). Understanding the diagnostics and etiology of diseases affecting *A. cervicornis*

Corresponding author
Margaret W. Miller,
margaret.w.miller@noaa.gov

populations remains problematic, and effective management strategies to combat this ongoing threat to species survival remain elusive. Despite more than a decade of focused research effort, there remains a dearth of strict diagnostic characterization for field cases of disease in *A. cervicornis* and, perhaps consequently, inconsistency in naming suspected disease conditions in published literature. Most authors simply apply the historical label of white-band disease (WBD) (*Aronson & Precht, 2001*; *Gignoux-Wolfsohn, Marks & Vollmer, 2012*; *Gladfelter et al., 1977*; *Peters, 1984*; *Vollmer & Kline, 2008*), a condition that was first described in *A. palmata* from Tague Bay, St. Croix, US Virgin Islands, as "a sharp line of advance where the distally located, brown zooxanthella-bearing coral tissue is cleanly and completely removed from the skeleton, leaving a sharp white zone about 1-cm wide that grades proximally into algal successional stages...". (illustrated in Fig. 1A; *Gladfelter, 1982*). *Peters, Oprandy & Yevich (1983)* found the same disease signs present on *A. cervicornis* colonies of the deeper forereef at Tague Bay (Table 1). A second type of WBD was recognized in the 1990s, WBD-II, distinguished by a section of bleached tissue at the tissue margin (Table 1; Fig. 1F; *Gil-Agudelo, Smith & Weil, 2006*; *Ritchie & Smith, 1998*).

The lesions attributed to WBD-I on Caribbean acroporids have varied in their patterns (smooth or ragged tissue margins) and rate (linear tissue loss less than 1 mm d$^{-1}$ to more than 14 mm d$^{-1}$, *Gladfelter, 1982*), and published descriptions have not always been clear (*Rogers, 2010*). For example, "rapidly advancing white band of diseased tissue" (*Vollmer & Kline, 2008*) is not appropriate because it is a band of white denuded skeleton, not white tissue, that appears progressively (does not itself advance) from the base or middle of a branch toward the branch tip as the necrotic tissue (confirmed by histological examination) peels off, sloughs, or lyses and disappears from the skeleton (*Gladfelter, 1982*; *Peters, Oprandy & Yevich, 1983*). In addition, recent observation of acute tissue loss in *A. cervicornis* in the Florida Keys indicates that lesions rarely present as a uniform-in-width band of denuded skeleton, as in the original description for WBD (quote above, Fig. 1C). Rather, initial lesions often show irregular sloughing of tissue with rapid enlargement of lesions anywhere on the surface of a branch, yielding multifocal swaths of bright white denuded skeleton. Due to lack of consistency with the original description (quoted above), some authors have refrained from using the name WBD-I in favor of the more general term rapid tissue loss (RTL) (Table 1, Fig. 1D, *Williams & Miller, 2005*). It should be noted that a similar but unnamed condition was described much earlier by *Bak & Criens (1981)* and that there is no evidence whether or not this condition is distinct from that referred to as WBD-I by other authors. Tissue loss on a colony can also appear as a combination of lesion types (Fig. 1E).

The ability to accurately identify disease in the field is further confounded by the activities of corallivores, such as the snail *Coralliophila abbreviata*, the polychaete *Hermodice carunculata*, and damselfishes or butterflyfishes, because they frequently remove *A. cervicornis* tissue and leave feeding scars that may be difficult to distinguish from disease (Table 1; *Bruckner, 2002a*; *Miller & Williams, 2006*; *Sutherland, Porter & Torres, 2004*). In addition, *Williams & Miller (2005)* found that *C. abbreviata* that were feeding at tissue-loss margins on disease-affected colonies could apparently transmit this condition

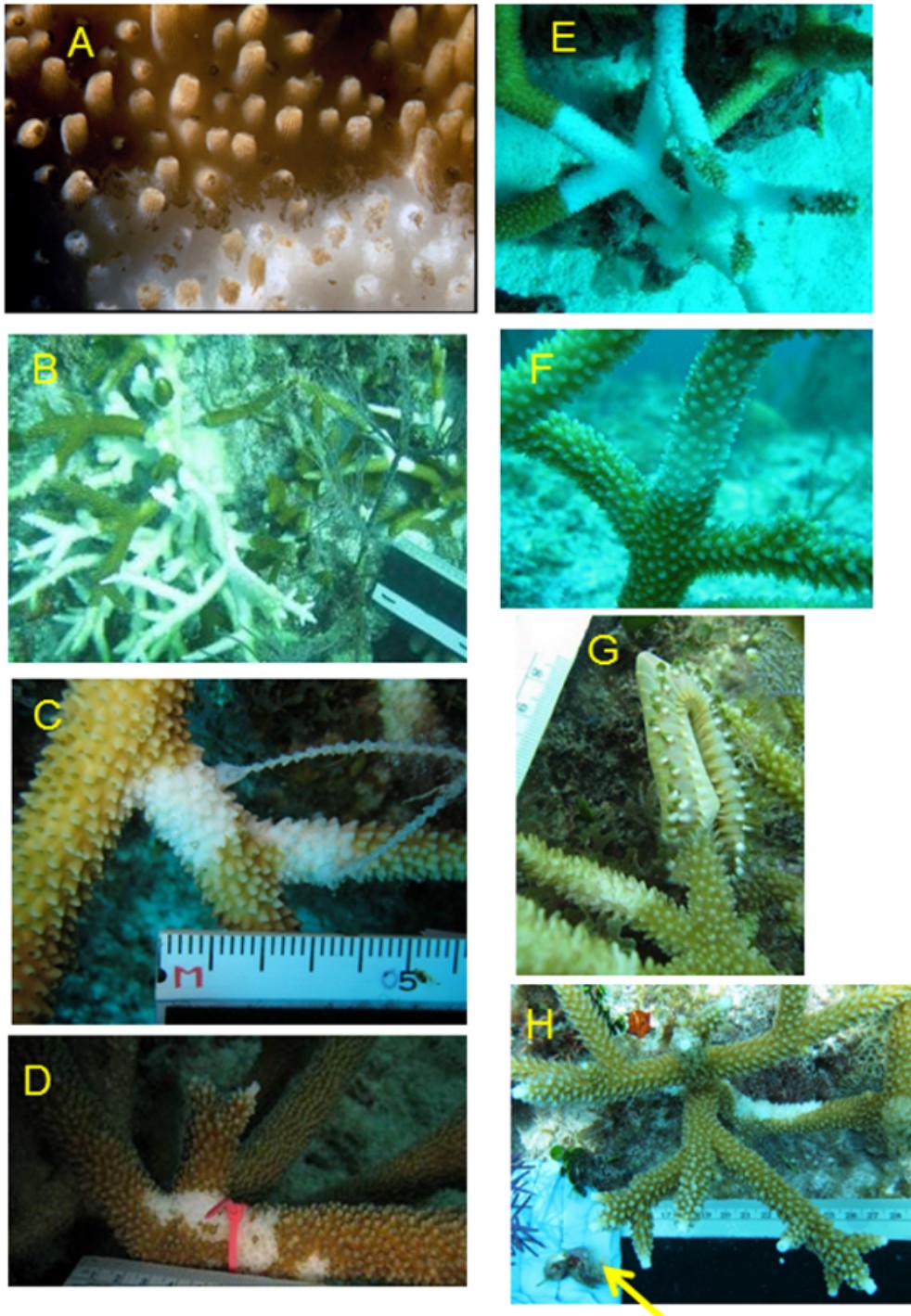

**Figure 1 Illustration of disease and predation conditions categorized in this study.** (A) Loss of necrotic tissue from skeleton of *A. palmata* during WBD outbreak, Tague Bay, St. Croix, 1980. (B) Typical disease-affected colony with diffuse lesions of denuded skeleton, (C) WBD-I, (D) initial stages of RTL, (E) colony manifesting signs of both WBD-I (base) and RTL (tips), (F) WBD-II signs, (G) fireworm predation with two older preyed tips (partially colonized by algal turfs) visible, and (H) snail predation scar on basal portion of branch (removed snails indicated by arrow).

**Table 1 Comparison of existing *Acropora cervicornis* disease descriptions.** Comparison of field manifestations of lesions seen in *A. cervicornis* and morphologic diagnoses. See *Work & Aeby (2006)* and *Galloway et al. (2007)* for definitions of terms.

| Field name | Tissue loss type | Location of lesion on colony* | Lesion margin appearance | Lesion shape and size | Lesion number and color | Lesion progression | Morphologic diagnosis |
|---|---|---|---|---|---|---|---|
| White-band disease type I (WBD-I)[a] | Acute to subacute | Base or middle of branch, encircling branch | Distinct areas of tissue loss, smooth to serpiginous margin, tissue tan to brown (due to symbiotic algae pigmentation) | Band of intact bare skeleton, well-differentiated from more distal skeleton | Focal to multifocal to diffuse, white (denuded skeleton), normally pigmented tissue margin | White band typically 2–10 cm wide; rate of tissue loss usually a few mm per day but can vary or stop; at branch bifurcation tissue loss continues on both branches at about the same rate; freshly denuded skeleton grades into green to brown algal growth on the skeleton, first visible after 5–7 days and becoming increasingly dense with time | Severe, basal to mid-branch band, diffuse, acute tissue loss, polyp, coenenchyme |
| White-band disease type II (WBD-II)[b] | Acute to subacute | Tip or base of branch, encircling branch | Distinct areas of tissue loss, smooth margin, 2–20 cm wide band of bleaching tissue (loss of brown algal pigmentation) between tissue loss margin and normally pigmented tissue | Band of intact bare skeleton, well-differentiated from more distal skeleton, developing green to brown algal growth | Focal to multifocal, white (denuded skeleton), bleaching tissue margin | White band typically 2–10 cm wide; rate of tissue loss usually a few mm per day; bleaching margin tissue disappears, normally pigmented tissue starts bleaching; however, bleaching margin tissue may also disappear and then the normally pigmented tissue disappears, as in WBD-I; freshly denuded skeleton grades into green to brown algal growth on the skeleton, first visible after 5–7 days and becoming increasingly dense with time | Severe, basal, band, diffuse, acute tissue loss, bleaching margin, polyp, coenenchyme |
| Rapid Tissue Loss (RTL)[c] | Acute | Basal, medial, or colony-wide, partially to completely encircling branch | Distinct areas of tissue loss, undulating to serpiginous margin, tan to brown tissue, sloughing | Irregularly shaped areas of intact bare skeleton | Focal to multifocal and coalescing to diffuse, white (denuded skeleton) | Intact bare skeleton appears quickly along branches, new lesions may coalesce; rate of tissue loss usually cm per day; denuded skeleton develops green to brown algal growth that becomes uniformly visible after 5–7 days covering entire denuded area | Severe, basal to complete, band or irregular, diffuse, acute to subacute tissue loss, polyp, coenenchyme |

Table 1 (*continued*)

| Field name | Tissue loss type | Location of lesion on colony[*] | Lesion margin appearance | Lesion shape and size | Lesion number and color | Lesion progression | Morphologic diagnosis |
|---|---|---|---|---|---|---|---|
| Fireworm (*H. carunculata*) predation feeding scars[a] | Acute | Apical 1–5 cm of branch, but not extending beyond a branch bifurcation | Distinct areas of tissue loss encircling apex of branch, smooth to serpiginous margins, tissue tan to brown | Intact bare skeleton, tip of branch, developing green to brown algal growth | Focal to diffuse, white (denuded skeleton) | None, denuded skeleton develops uniform green to brown algal growth | Severe, focal, branch tip, acute tissue loss, polyp, coenenchyme |
| Snail (*C. abbreviata*) predation feeding scars[a] | Acute | Colony base, skeletal-tissue margin inward and vertically | Distinct areas of tissue loss, smooth to serpiginous rounded or scalloped margins, tissue tan to brown | Intact bare skeleton, usually adjacent to one or more *Coralliophila abbreviata* | Focal or multifocal, white (denuded skeleton) | None, denuded skeleton gradually colonized by green to brown algal growth | Severe, diffuse, basal, acute tissue loss, polyp, coenenchyme |

**Notes.**

[*] First lesion on all of these may be a single small focus of acute tissue loss, either at the base or in the middle of a branch, lesion enlargement pattern then varies.

[a] Illustrated in *Williams, Miller & Kramer (2006)* but only for *A. palmata*.

[b] Described in *Ritchie & Smith (1998)*.

[c] Described in *Williams & Miller (2005)*; described but not named in *Bak & Criens (1981)*.

when subsequently allowed to feed on apparently healthy branches, resulting in continued tissue loss; thus, predation may exacerbate disease spread through a population.

*Acropora cervicornis*' status under the USA Endangered Species Act carries a legal mandate to orchestrate its recovery (i.e., a sustainable status where ESA protections are no longer needed to prevent extinction). This mandate, combined with a growing consensus that decline has reached a point where natural resilience is likely compromised, has led to increasing efforts to culture and restock populations of *A. cervicornis* (reviewed in *Young, Schopmeyer & Lirman, 2012*). This unprecedented movement toward proactive intervention and population engineering in a coral reef foundation species is occurring within a historical context of mixed success in previous case studies in the fields of fisheries and wildlife management (*Carlsson et al., 2008*; *Champagnon et al., 2012*; *Hilborn & Eggers, 2000*). The primary concern for such an endeavor is the potential for unintended introductions of deleterious genetic or health consequences within the imperiled population or its ecosystem (*Baums, 2008*; *Cunningham, 1996*). For this reason, the genetic status of imperiled coral populations, including *A. cervicornis*, has received increasing attention in recent years and strides have been made in addressing the potential genetic risks of culturing and restoring *A. cervicornis* populations, such as outbreeding depression or genetic bottlenecks in cultured stocks (*Baums et al., 2010*; *Hemond & Vollmer, 2010*).

Addressing potential health risks of transplanting *Acropora cervicornis*, on the other hand, is much more challenging. While explicit risk assessment and risk management frameworks have been proposed and applied in wildlife translocation projects, effective application requires at least qualitative knowledge of pathogens, vectors, and susceptibilities operating in the given species (e.g., *Lenihan et al., 1999*; *Sainsbury & Vaughan-Higgins, 2012*). The limited use of multidisciplinary effective diagnostic tools and lack of robust etiological characterization for coral disease in general, and in *A. cervicornis* in particular (*Rogers, 2010*; *Sutherland, Porter & Torres, 2004*), impairs efficient health risk management. Until a better knowledge base is built for health management of coral populations, presumed risk-averse 'best practices' are currently applied in nursery culture and outplanting (or restocking) of *A. cervicornis*. These practices include emphasis on field-based (rather than land-based) culture, avoiding outplanting colonies with visual signs of ill health (discoloration or tissue loss), geographic matching of source populations, nursery sites, and target sites, and targeting outplants at sites where there is evidence of prior occupation, but without extant live wild colonies (*Johnson et al., 2011*; L Gregg, Florida Wildlife Conservation Commission, pers. comm., 2011).

The severe and ongoing impact of coral disease on coral populations begs the question of potential mitigation actions that could be applied in the context of local management (*Beeden et al., 2012*; *Bruckner, 2002b*; *Raymundo, Couch & Harvell, 2008*). If effective, such targeted mitigation actions would seem particularly relevant and useful as part of an integrated health-risk management component in a population restocking program. Both nursery and field practitioners have anecdotally reported simple interventions, such as separating apparently healthy tissues from diseased colonies or applying a physical barrier (e.g., band of clay or epoxy) to the diseased tissue margin to control tissue loss

(*Johnson et al., 2011*; *Raymundo, Couch & Harvell, 2008*), but no controlled tests of such mitigation treatments have been published. Indeed, we are aware of only two published studies reporting on successful field disease mitigation treatments; 80% successful excision of distal *Turbinaria* spp. white-syndrome lesions (*Dalton et al., 2010*), and anecdotal success of aspirating black-band microbial mats with subsequent clay seal over the tissue margin (*Hudson, 2000*). However, neither study examined the treated colonies' tissues microscopically to determine why their treatments were successful.

The objectives of the present study were to: (1) characterize disease dynamics using targeted disease surveillance in outplanted/transplanted versus wild populations of *A. cervicornis* to provide a more robust scientific basis for judging the health risks associated with outplanting, and (2) perform controlled tests of two simple mitigation treatments *in situ* to determine if they significantly arrested tissue loss in affected colonies. For both objectives, and to improve our understanding of the tissue loss diseases in this species, the histopathology of selected fragments from unmanipulated and treated branches was evaluated using light microscopy.

## MATERIALS AND METHODS

### Study sites

Disease prevalence surveys and mitigation treatments were conducted at restored and wild *A. cervicornis* populations in the upper Florida Keys National Marine Sanctuary. Restored populations were outplanted between 2007 and 2011 as part of previous projects by the Coral Restoration Foundation (CRF) or the National Marine Fisheries Service-Southeast Fisheries Science Center. Each restored site hosted either outplanted (i.e., from field nursery culture) or transplanted (i.e., from nearby wild populations) colonies, with one site (Aquarius) hosting a mixture of both sources of restored colonies (Table 2). These restored sites were deliberately established in areas devoid of native wild colonies and are in shallow (3–8 m) fore-reef habitats, including Key Largo Dry Rocks, French, Molasses, Pickles, and Conch Shallow reefs (Table 2; Fig. S1). An additional restored site (Aquarius) was surveyed in 2011 only and was located in the deeper fore-reef (14–16 m) of Conch Reef. Few wild *A. cervicornis* patches are extant in the upper Florida Keys; three were identified for the current study to provide comparison to the restored populations. These wild sites were all located in low-relief patch reefs with partially consolidated rubble bottom at about 5-m depth and included an unnamed patch reef off of Tavernier, FL (TavPatch sites A and B), and Little Conch reef. Periodic surveys were also conducted at the CRF field nursery (origin of most restored colonies).

Temperature data were collected at surveyed reefs during the survey seasons with HOBO pendant data loggers (UA-001-64; Onset Corporation). Loggers were not re-located at TavPatch or Key Largo Dry Rocks after Tropical Storm Isaac so temperature data for those two sites are not available in 2012.

The study was conducted under Florida Keys National Marine Sanctuary Permit #FKNMS-2011-032-A1.

**Table 2 Characteristics of study sites/populations and mitigation experiment in the upper Florida keys.** Number of genets indicates number of *Acropora cervicornis* multi-locus genotypes (based on seven microsatellite markers (*Baums et al., 2009*; *Baums et al., 2010*) within the surveyed populations at each site. Distribution of experimental replicates for the mitigation experiment among sites and years is summarized in the last two columns.

| | Colony origin | Site type | # of genets | Coordinates | Depth (m) | # 2011 replicates (C/EB/EX) | # 2012 replicates (C/EB/EX) |
|---|---|---|---|---|---|---|---|
| Molasses | Nursery | Restored | 3 | 25°00.60′N 80°22.37′W | 8–10 | 2/8/6 | 1/1/0 |
| Aquarius | Transplant & nursery | Restored | 11 | 24°57.20′N 80°27.15′W | 14 | 9/6/3 | NA |
| Conch Shallow | Transplant | Restored | 14 | 24°57.08′N 80°27.59′W | 6 | 1/1/1 | 4/4/5 |
| French | Nursery | Restored | 3 | 25°07.31′N 80°17.85′W | 10 | 5/3/4 | 6/5/1 |
| KL Dry Rocks | Nursery | Restored | 3 | 25°07.45′N 80°17.84′W | 6 | NA | 3/3/4 |
| Pickles | Nursery | Restored | 3 | 24°59.30′N 80°24.74′W | 8–10 | 0/1/1 | NA |
| Tav patch A | Wild | Wild | UNK[*] | 24°59.23′N 80°27.17′W | 6 | NA | NA |
| Tav patch B | Wild | Wild | UNK | 24°59.24′N 80°27.16′W | 6 | NA | NA |
| Little Conch | Wild | Wild | UNK | 24°56.78′N 80°28.21′W | 6 | NA | 10/10/2 |
| CRF nursery | | Nursery | >20 | 24°59′N 80°26′W | 11 | NA | NA |

**Notes.**
[*] Previous haphazard genotype sampling at this site yielded 6 unique genets in 20 sampled colonies (MW Miller & IB Baums, 2008, unpublished data).
UNK, Unknown; C, Control; EB, Epoxy band; EX, Excision.

## Surveillance

Disease surveillance was conducted from May to November in 2011 and 2012 to target the seasonal time frame when acroporid disease was expected to be most active (*Williams & Miller, 2005*; K Nedimyer, pers. comm., 2004). Surveys were conducted approximately every two weeks in 2011 (total nine surveys) and monthly in 2012 (total seven surveys), each taking 2–3 days to complete. At each wild site, a fixed circular plot was marked with a center rebar stake and used to delineate the study population for which prevalence was determined (i.e., percent of colonies in the population that displayed signs of disease). Different plot sizes (8-m radius at Tav Patch A and B, 10-m radius at Little Conch) were used at the wild sites to incorporate a minimum of 25 colonies. At restored sites, the sample population consisted of the outplanted and/or transplanted colonies present. The number of colonies tallied for individual site prevalence estimates ranged from 23 to 163 according to the number of colonies available and the extent of search during a given survey.

During each survey, every colony was recorded as either affected or unaffected with acute tissue loss disease including both WBD and RTL descriptions (Table 1; i.e., bright white skeleton with either a straight or jagged tissue margin on basal or interstitial portions of the colony or multifocal). Corallivory was also common, so basal lesions with snails

present or denuded branch tips not passing a fork which are characteristic of fireworm feeding (Table 1; Fig. 1G; *Shinn, 1976*) were not counted as disease. Prevalence was calculated for each site for every survey and averaged for each site-by-year combination. A two-way, fixed-factor ANOVA, with factors being site-type (restored versus wild) and year (2011 or 2012) and sites as replicates, was conducted to determine if overall prevalence varied significantly between restored and wild sites or years. In addition, as a qualitative comparison, disease prevalence observations were also made during six surveys in 2011 and one in 2012 at the nearby field nursery (Coral Restoration Foundation) from which all the outplanted colonies in the study had originated.

To characterize disease incidence and mortality, 20 haphazardly selected colonies were tagged at each site in May 2012. At each survey, tagged colonies were photographed and a visual estimate of percent of dead colony surface (in 5% increments), attributed as either predation, disease, or undefined, was recorded. After the fifth survey, disturbance from Tropical Storm Isaac damaged or removed several tagged colonies at most sites, resulting in observations of fewer than 20 colonies at the sixth survey. To determine disease incidence (rate of new disease cases) over a survey interval, each colony observed with active disease that had been observed as unaffected at the previous survey was counted as a new disease case. Incidence was expressed as a proportion of observed tagged colonies displaying new cases of disease since the previous survey and was standardized per week. Separate *t*-tests were used to determine if (1) incidence averaged over time and (2) the proportion of tagged colonies that remained unaffected during 2012, differed between restored and wild sites.

We estimated partial mortality based on cumulative increase in rough visual estimates of percent dead on each of the tagged colonies that was observed with disease. To help discern the effect of Tropical Storm Isaac, we analyzed cumulative partial mortality for all cases that occurred prior to the storm (through survey five), and then including new cases that were observed at the survey following the storm (survey six). A *z*-test was used to compare the proportion of affected wild vs. affected restored colonies showing severe cumulative partial mortality (defined as greater than 80%). We also tallied the case fatality rate as the percent of cases (i.e., colonies that displayed disease signs during the course of the observation period) undergoing complete mortality.

## Mitigation experiment

Two disease mitigation treatments were implemented to test effectiveness in arresting tissue loss (Fig. 2). The first treatment used a band of two-part marine epoxy (All-Fix Epoxy) applied around the branch to cover the disease margin of an affected colony, presumably functioning as a physical barrier over the tissue-loss margin. The second treatment involved a complete excision of live, apparently healthy, tips of branches distal to a disease margin using handheld wire cutters. The excised fragment was then reattached to the reef substrate with epoxy at a distance greater than 1 m from the parent colony. These treatments are referred to as excision (EX; Fig. 2A) and epoxy band (EB; Figs. 2B–2C), respectively. Lastly, a control treatment consisted of a cable tie placed at or near a tissue loss

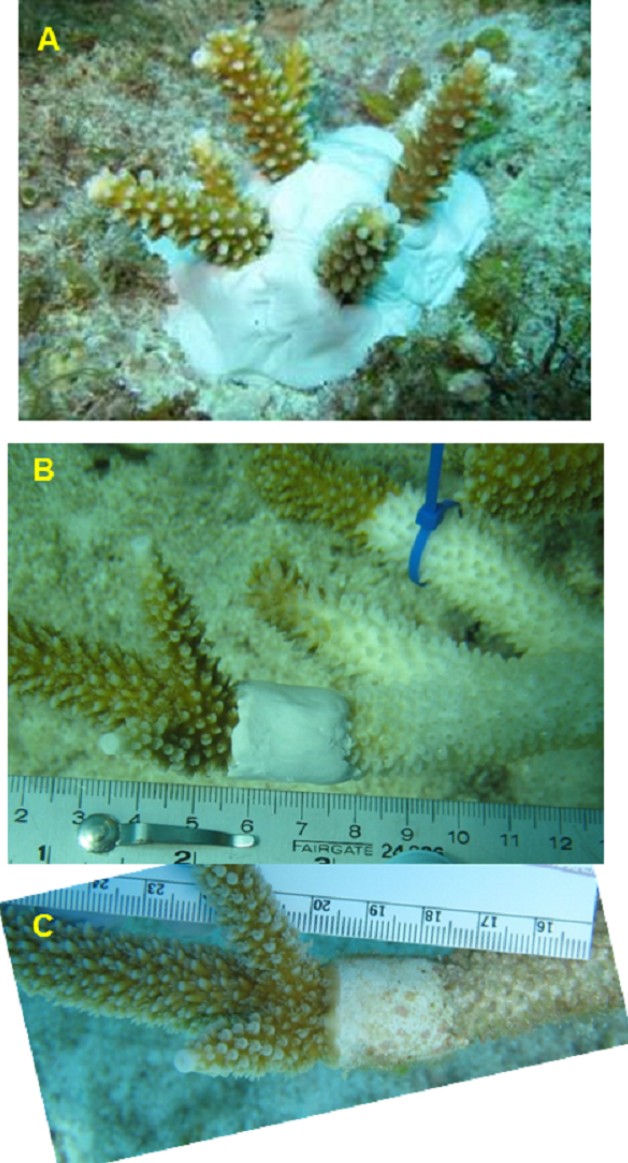

**Figure 2 Illustration of the treatments used in mitigation trials.** (A) Excision (EX) of healthy looking tips snipped from a nearby disease colony and re-attached to the reef, (B) Epoxy band (EB) surrounding the diseased tissue margin. One month later (C) this 'successful' EB replicate shows no additional tissue loss and initial regrowth over the epoxy. Control treatments are illustrated in Figs. 1C and 1D.

margin on the same colony as a reference point to detect continued tissue loss (Fig. 1C or Fig. 1D). To prevent potential contamination, nitrile gloves were used when manipulating colonies and were changed when moving between affected colonies. All equipment that came into contact with diseased colonies was rinsed in a 10% bleach solution following each dive.

The design and setup for this experiment, including sample size, timing, and placement of replicates, were constrained by the availability of affected colonies with apparently

active disease. Due to permitting constraints in 2011, no experimental mitigation treatments were performed on wild *A. cervicornis* colonies. In 2012, this stricture was lifted and treatments were conducted on both restored and wild colonies. Distribution of experimental replicates among sites and years is given in Table 2. Effort was taken to block treatments within the same colony if it contained three or four (to include a histology sample) affected branches. However, this was often not possible and so treatments were allocated sequentially to affected colonies as they were encountered.

Rates of tissue loss in the observed diseased conditions were rapid so all experimental replicates were scored as either (1) continued or (2) arrested tissue loss at an interval of approximately one month after the treatment was implemented, and each treated colony was photographed to document tissue loss progression. In some cases, corallivores were subsequently observed on a treated or control colony. These replicates were excluded from analysis because continued tissue loss could not be confidently attributed to disease and hence would not confidently constitute 'failure' of the treatment. Proportion of replicates with continued versus no tissue loss after applying the treatment was compared among the three treatments using Chi-Squared tests (for each year separately and for the years pooled).

## Histopathology

To better characterize the observed disease conditions, tissue samples were collected in 2011 from a subset of apparently healthy colonies ($n = 21$, including at least one and up to four colonies from each site, collected in June or late September 2011), diseased colonies observed in the vicinity of the surveys ($n = 12$), and diseased samples collected from the colonies in the mitigation experiment ($n = 11$) collected throughout the sampling season. In addition, two diseased samples were collected from wild site Little Conch in 2012 to compare with the apparently healthy samples collected at that site in 2011. Samples were removed by cutting a 5–10 cm portion of a branch including tissue and skeleton, using handheld wire cutters and placed in a labeled 50-ml plastic centrifuge tube. After surfacing, the sample was immediately immersed in a formaldehyde-based fixative solution (Z-Fix Concentrate, Anatech, Ltd., 1:4 dilution in seawater). Sample tubes were capped, kept at ambient temperature in the shade, and shipped to the Histology Laboratory at George Mason University for processing.

Each sample was photographed and the images compiled into trim sheets. Samples were trimmed into approximately 2-cm long fragments using a Dremel tool and diamond-coated tile-cutting blade. The location of each cut was marked on the sample image on the trim sheet and subsample numbers were assigned and marked on the trim sheet. Subsamples having a tissue loss margin were enrobed in 1.5% agarose to trap material that might be present on the denuded skeletal surface or in corallite or gastrovascular canal crevices. Subsamples were decalcified using 10% disodium ethylenediaminetetraacetic acid (EDTA) at pH 7, changing the solution every 24–48 h. When completely decalcified, the subsamples were rinsed in running tap water for about 30 min, trimmed into 2–3 mm slices and placed in cassettes, processed through ethanols, cleared, and infiltrated with molten Paraplast Plus®, then embedded in Paraplast Xtra®

(*Peters, Price & Borsay Horowitz, 2005*). Sections (5-μm thickness) were mounted on microscope slides, stained with Harris's hematoxylin and eosin and Giemsa (*Noguchi, 1926*) procedures, and coverslipped with Permount™ mounting medium.

The sections were examined with an Olympus BX43 compound microscope and photomicrographs obtained with an Olympus DP-72 camera. Semi-quantitative data (*Jagoe, 1996*) were collected from each subsample based on relative condition (tissue architecture, cellular integrity, zooxanthellae abundance, pathological changes) at the time of fixation (0 = Excellent, 1 = Very Good, 2 = Good, 3 = Fair, 4 = Poor, 5 = Very Poor) and severity or intensity of tissue changes from normal (0 = Within Normal Limits, 1 = Minimal, 2 = Mild, 3 = Moderate, 4 = Marked, 5 = Severe) (see Table S1). Histoslides of *A. cervicornis* and *A. palmata* collected from the 1970s in the Florida Keys (the earliest tissue samples located, before tissue loss was reported from this region) were used to develop the "within normal limits" criteria for general coral tissue condition and zooxanthellae condition/abundance scores, six specific cell or tissue parameters of polyp health, bacterial aggregates (*Peters, Oprandy & Yevich, 1983*), and suspect rickettsia-like organisms (RLOs) (*Casas et al., 2004*; CS Friedman, pers. comm., 2010). Presence/absence was noted for hypertrophied calicodermis foci, necrotic cell spherules, apicomplexans (*Upton & Peters, 1986*), and suspect ciliate predators. The developmental stage of gonads was noted, if present (*Szmant, 1986*). Mean scores for each sample were obtained (one or multiple sections were made, especially if enrobed samples had been trimmed into four ∼2–3 mm slices for embedding; some sections did not contain enough tissue for scoring) and checked for quality. Suspect RLO abundances were visibly higher in Giemsa-stained sections since it demonstrates *Rickettsia* well (*Noguchi, 1926*); thus, estimates based on those sections were preferentially used. Descriptive statistics were calculated for the scored parameters in each group of samples (apparently healthy, disease characterization, and mitigation treatments). Frequency distributions of the scores were examined. Comparisons were made for the scored parameters between all apparently healthy and diseased samples, successful and unsuccessful mitigation treatments, and WBD- and RTL-affected samples using Student's *t*-tests and Mann–Whitney *U*-tests.

## RESULTS

### Disease dynamics

Intermittent observations within the field nursery throughout 2011 yielded consistently low prevalence of 0–1.7%. However, disease prevalence in reef populations was highly variable and largely site-specific with no consistent patterns between restored versus wild sites (Figs. 3A–3D). In 2011, wild sites showed relatively low prevalence with means of 1.5 to 4.4% during the survey period and a peak of approximately 13% at TavPatch B in late June (Fig. 3A). Meanwhile, four of six restored sites showed generally high disease prevalence (i.e., survey period means of 9–17% and max of 26–41%; Fig. 3C) particularly from July through early October, while the remaining two restored sites showed consistently low prevalence throughout 2011 (i.e., Key Largo Dry Rocks and Conch Shallow had 2011 survey period means of 0.7 and 3.5% prevalence with one peak of 13%, lower or similar to

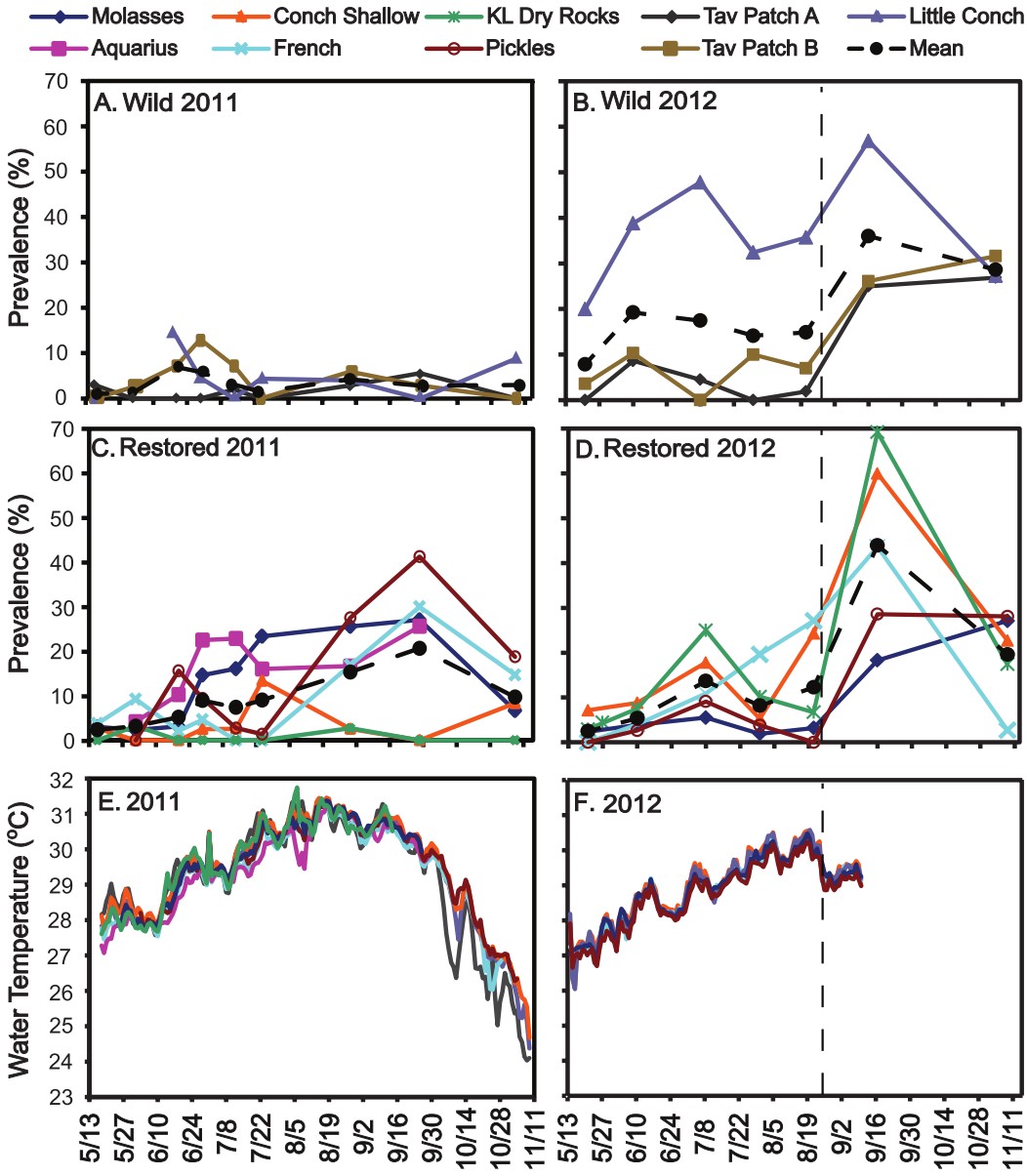

**Figure 3 Disease prevalence and temperature.** Disease prevalence in *Acropora cervicornis* colonies in Wild (A and B) and Restored (C and D) populations over two survey periods (May–Nov 2011 and May–Nov 2012). Dotted lines indicate close passage of Tropical Storm Isaac in Aug 2012. (E) and (F) show the temperature records from the same sites and time periods.

the wild sites; Fig. 3C). In contrast, during 2012, Key Largo Dry Rocks and Conch Shallow showed among the highest prevalence patterns with survey period means of 20% and peaks of 60–70% (Fig. 3C). Little Conch (wild) consistently had the highest site prevalence throughout the 2012 survey period (20–57% range, mean 35%; Fig. 3B).

The temperature records indicate little temperature variation among sites during both years (Figs. 3E and 3F), suggesting that site-specific disease increases or outbreaks were not triggered by temperature. Additionally, the accumulated thermal stress (i.e., the cumulative

**Table 3 Disease incidence in 2012.** Survey intervals (dates and duration in weeks), incidence, and proportion of colonies that remained unaffected by disease for the population of tagged colonies ($n = 20$) at each site throughout the 2012 sampling period. Incidence is expressed as the proportion of new cases observed during each survey interval (i.e., diseased tagged colonies observed without disease in the previous survey) standardized per week. Shading is (arbitrarily) scaled with incidence value with gradually darker shading indicating higher incidence (cutoff levels of 0.01, 0.025, 0.05, 0.1 and 0.15). Tropical Storm Isaac passed during Interval V.

| | Interval | I | II | III | IV | V | Unaffected |
| --- | --- | --- | --- | --- | --- | --- | --- |
| | Dates | 5/15–6/2 | 6/2–6/30 | 6/30–7/23 | 7/23–8/15 | 8/15–9/10 | |
| | (#weeks) | (2.71) | (4.00) | (3.29) | (3.29) | (3.71) | |
| Restored | Conch shallow | 0.018 | 0.000 | 0.000 | 0.064 | 0.126 | 0.400 |
| | Pickles | 0.000 | 0.025 | 0.000 | 0.000 | 0.094 | 0.400 |
| | Molasses | 0.037 | 0.025 | 0.000 | 0.016 | 0.000 | 0.800 |
| | French | 0.000 | 0.050 | 0.046 | 0.076 | 0.075 | 0.250 |
| | KL dry rocks | 0.018 | 0.088 | 0.000 | 0.015 | 0.184 | 0.150 |
| Wild | Little conch | 0.037 | 0.063 | 0.046 | 0.076 | 0.099 | 0.000 |
| | Tav patch A | 0.018 | 0.013 | 0.000 | 0.000 | 0.038 | 0.800 |
| | Tav patch B | 0.018 | 0.000 | 0.016 | 0.015 | 0.058 | 0.700 |

duration of temperature exposure >30 °C) was greater in 2011 than in 2012 (Figs. 3E and 3F), but this did not correspond to higher disease prevalence. The mean prevalence during the survey period was higher in all three wild sites and four of six restored sites in 2012 than 2011. In contrast, the passage of Tropical Storm Isaac (26 Aug 2012) did correspond to a ubiquitous spike in disease prevalence across all sites (restored and wild). A two-way ANOVA using site means for each year showed a significant effect of year ($p = 0.032$) but not of site-type ($p = 0.786$) nor the interaction ($p = 0.237$). However, if the post-storm prevalence surveys are excluded in 2012, no factors are significant, suggesting that higher overall disease prevalence in 2012 was attributable to the acute effect of the storm.

Temporal patterns of disease incidence in 2012 are shown in Table 3 and further emphasize the site-specific nature of disease dynamics in this population. Individual sites show widely varying patterns of incidence, from persistent low incidence followed by a spike in the fifth interval, following Tropical Storm Isaac (e.g., Pickles, TavPatch-A, TavPatch-B), to a moderate level in the first three intervals followed by declining incidence (Molasses), to sites with persistently high incidence from interval two (French, Little Conch), to sites with both an early and a late peak (intervals two and six; Key Largo Dry Rocks) (Table 3). Average incidence did not differ significantly between Restored and Wild sites (2 sample $t$-test; $t = 0.323$, 6 degrees of freedom, $p = 0.757$) though this test has very low power (0.05).

Among the initial tagged population of 160 colonies in 2012, a total of 89 disease cases were identified with a case fatality rate of 7.9%. The proportion of colonies that remained unaffected throughout the study (non-cases, Table 3) was not significantly different between restored and wild sites ($t$-test, $p = 0.686$). Prior to the storm (up to survey 5), only $n = 53$ cases occurred. Fifty-two % of these cases showed no detectable increment of partial mortality (Fig. 4) and there were similar frequencies of cumulative partial mortality between restored and wild cases. When the storm interval is included, disease-affected

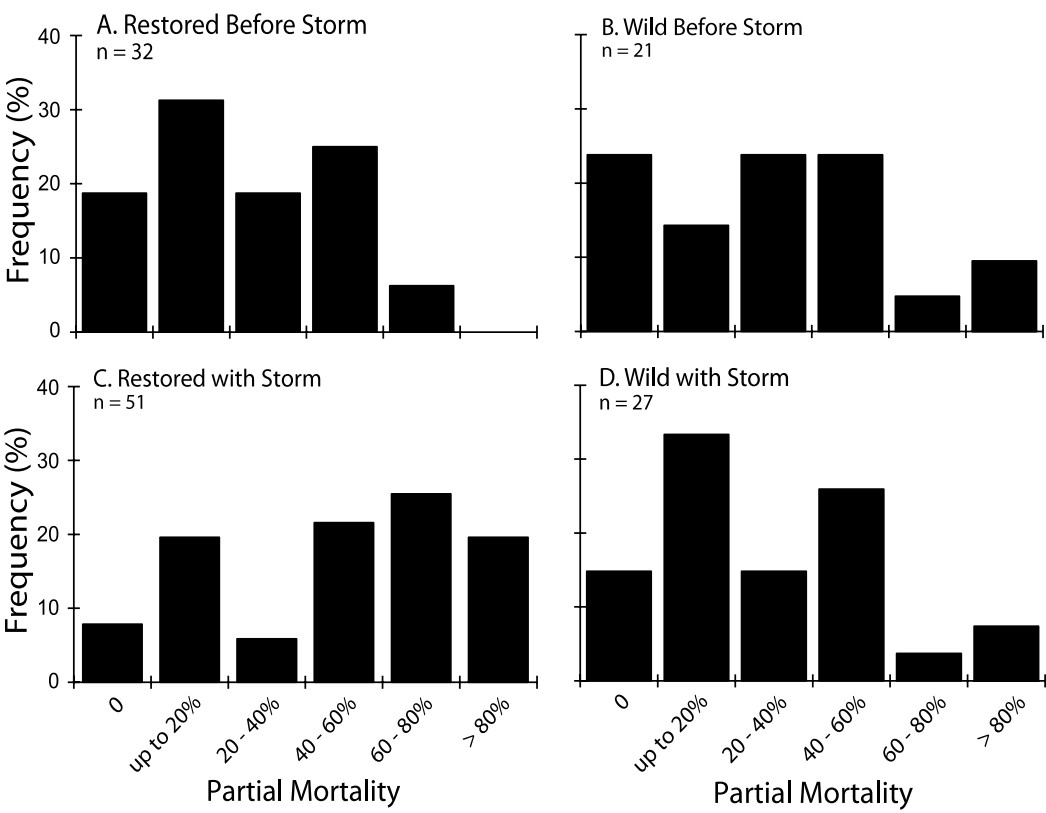

**Figure 4 Partial mortality in diseased colonies.** Frequencies of cumulative partial mortality in tagged diseased colonies during the 2012 survey period before (A and B, Surveys 1–5) and after (C and D, Surveys 1–6) passage of Tropical Storm Isaac at Restored and Wild sites. The bin labeled zero includes colonies that were observed with disease signs but accumulated less partial mortality than could be resolved in coarse visual estimates.

restored colonies had a significantly greater likelihood of showing severe (>80%) partial mortality than affected wild colonies (Fig. 4; $z$-test, $p = 0.005$).

## Mitigation experiment

Approximately 60–70% of control replicates in each year showed continued tissue loss after one month (Fig. 5). In other words, around one-third of the replicates we thought to be in an active diseased state based on gross visual signs were, in fact, dormant during the following one-month period of observation. The proportion of experimental replicates displaying tissue loss about one month after the treatment application did not differ significantly among EB, EX, and Control treatments for either year analyzed separately (2011: $\chi^2 = 0.134, p = 0.935$; 2012: $\chi^2 = 1.502, p = 0.472$) nor for both years pooled ($\chi^2 = 0.953, p = 0.621$). However, the power of these tests is very low (0.059–0.173) so negative results should be treated with caution.

## Histopathological observations

Summary statistics for the apparently healthy samples collected during the surveys, diseased samples for characterization, and diseased mitigation samples are presented in

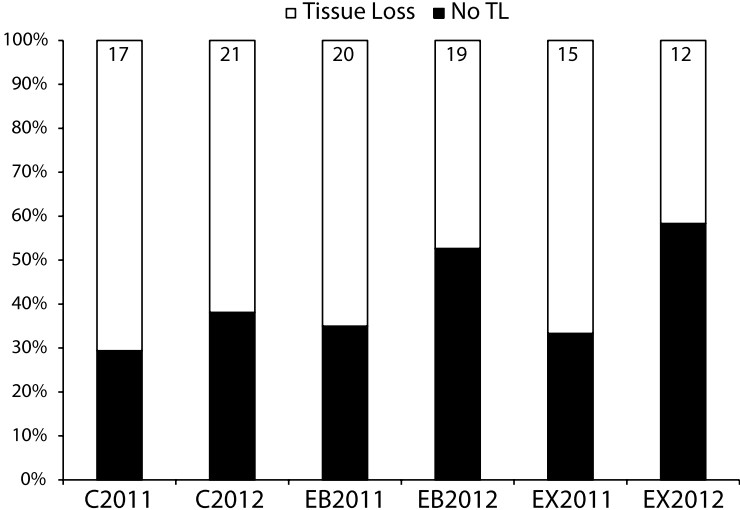

**Figure 5 Mitigation experiment.** Results of experimental mitigation trials showing response in each year for Epoxy-Band (EB), Excision (EX) and Control (C, cable tie placed around disease margin on a branch) treatments as the percent of replicates showing continued tissue loss after one month. Number of replicates is given above each bar. Chi-Squared Goodness of Fit tests indicate no significant difference in the proportions of the three treatments showing continued tissue loss when all replicates across years are pooled.

Table 4. The apparently healthy samples were in very good to fair condition, had more zooxanthellae in gastrodermal cells, numerous mucocytes filled with pale, frothy mucus that were about the same height as the ciliated columnar cells of the epidermis (Fig. 6A), and intact cnidoglandular bands of the mesenterial filaments (Fig. 6B). A third of the samples had minimal gaps (complete loss of tissue) in the calicodermis, mesoglea, and epidermis of the surface body wall covering costal ridges on the outside of the polyp's theca. The calicodermis toward branch surfaces was squamous to columnar, relatively thick, and contiguous over the mesoglea; calicoblasts often showed plasmallema extensions on their apical surfaces (toward the skeleton) and pale pink to clear cytoplasm (Fig. 6C). Deeper calicodermis was squamous and the cytoplasm contained fine eosinophilic granules. None of the samples contained bacterial aggregates, but almost all had mild to marked numbers of suspect RLOs, identified as basophilic clusters of large pleomorphic to uniformly coccoid cells staining red to purple with Giemsa, in mucocytes on polyp oral discs and tentacles (Fig. 6D) and in cnidoglandular bands of the mesenterial filaments (Fig. 6E). Coccidian oocysts were seen in two samples. Early oocytes were found in two samples, but no spermaries were observed.

Generally, characteristics of the diseased tissue samples collected from restored colonies at a range of sites throughout the 2011 season included moderate to severe attenuation of the epithelia and mesoglea, numerous hypertrophied mucocytes or reduced number of mucocytes in the epidermis (Fig. 6F), and reduced numbers of zooxanthellae (but not entirely missing). Cells of the cnidoglandular bands showed varying degrees of atrophy, loss, necrosis or apoptosis, and dissociation (Fig. 6G). Moderate to severe costal tissue loss was noted, beginning in the apical polyp and increasing toward the tissue loss

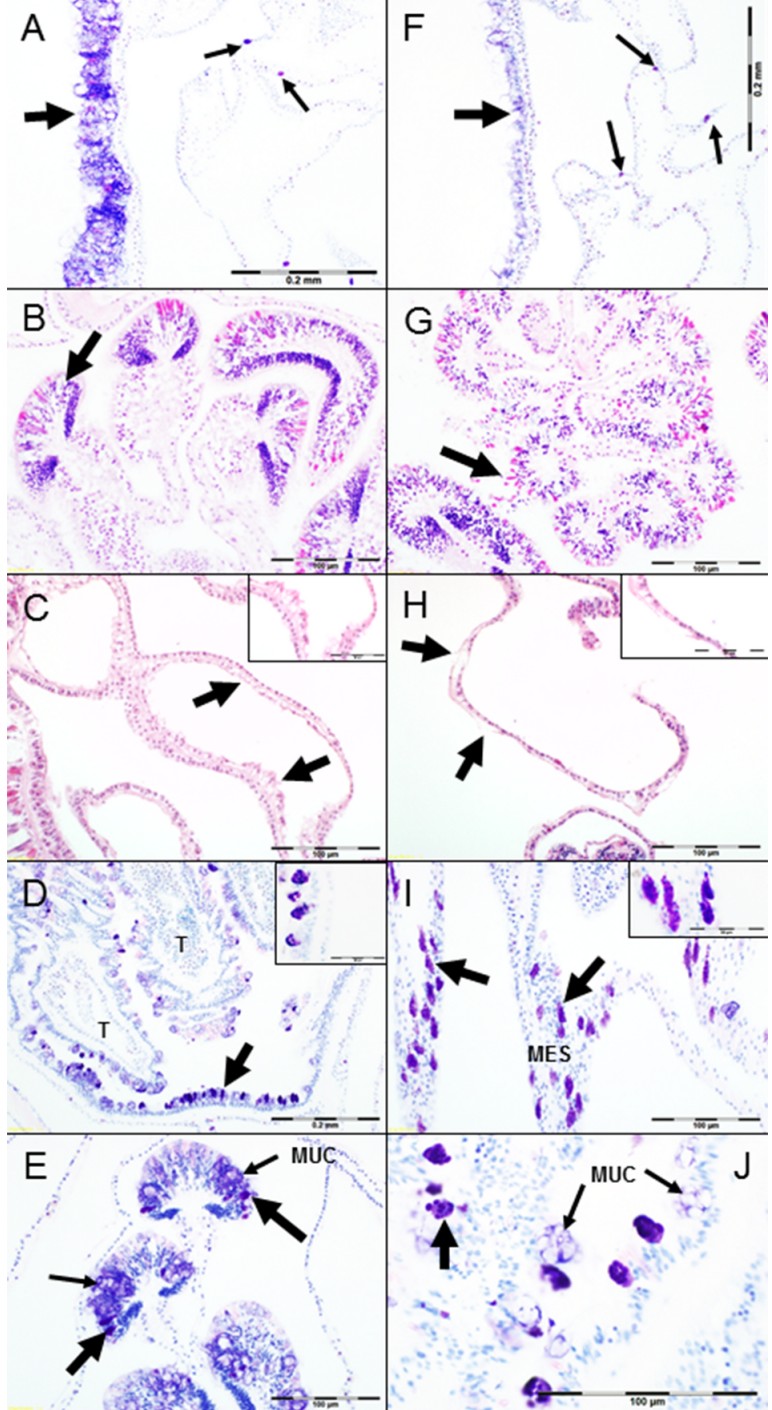

**Figure 6** **Histology observations.** (A) Coenenchyme epidermis from apparently healthy *Acropora cervicornis* branch tip, columnar mucocytes of surface body wall (large arrow), suspect RLOs in gastrodermal mucocytes of basal body wall (small arrows), Giemsa. (B) Mesenteries showing sections through cnidoglandular bands (large arrow), (H & E). (C) Apparently healthy staghorn sample, epithelia lining gastrovascular canals with columnar calicoblasts having extensions of plasmallema (large arrows (continued on next page...)

**Figure 6 (...continued)**

and inset), (H & E). (D) Section through tentacles (=T) and oral disc from apparently healthy colony sample, mucocytes infected with suspect RLOs stain dark purple (large arrow pointing to oral disc and inset), Giemsa. (E) Cnidoglandular bands from apparently healthy colony sample, suspect RLOs in mucocytes (large arrows) and mucocytes in the epithelium (small arrows). (F) Coenenchyme epidermis from *A. cervicornis* showing signs of RTL, note atrophy of epithelium and loss of mucocytes (large arrow), suspect RLOs in gastrodermal mucocytes of basal body wall (small arrows), Giemsa. (G) Sections through mesenteries from RTL-affected sample with degeneration (necrosis, lysing) and dissociation of cells of the cnidoglandular bands, note pink-staining acidophilic granular gland cells are rounding up and atrophied, ciliated cells and mucocytes are reduced in number compared to Fig. 6B, (H & E). (H) RTL-affected sample epithelia lining gastrovascular canals, severe atrophy of the calicodermis, loss of calicoblasts from mesoglea (large arrows and inset); adjacent gastrodermis is swollen, fragmented, and vacuolated compared to cuboidal cells in upper left corner of image, (H & E). (I) Suspect RLOs infecting gastrodermal cells (large arrows and inset) lining the mesenteries (=MES) of an apparently healthy sample, Giemsa. (J) High magnification of infected epidermal mucocytes from apparently healthy sample, showing pleomorphic suspect RLOs (large arrow) and mucocytes (small arrows, = MUC), Giemsa.

**Table 4  Histopathology summary.** Summary statistics for histopathological observations on all apparently healthy ($n = 21$), diseased ($n = 11$), and mitigation treatment samples ($n = 11$). Condition scores applied were 0 = Excellent, 1 = Very Good, 2 = Good, 3 = Fair, 4 = Poor, 5 = Very Poor; scoring of severity or intensity of tissue changes from normal were 0 = Within Normal Limits, 1 = Minimal, 2 = Mild, 3 = Moderate, 4 = Marked, 5 = Severe.

| Parameter | Apparently healthy | | | Characterization diseased | | | Mitigation treatments | | |
|---|---|---|---|---|---|---|---|---|---|
| *Assigned scores* | Mean | St.Dev. | Range | Mean | St.Dev. | Range | Mean | St.Dev. | Range |
| General condition (100×) | 1.6 | 0.7 | 1–3 | 4.5 | 0.5 | 3–5 | 4.4 | 0.7 | 3–5 |
| Zooxanthellae (100×) | 1.2 | 0.5 | 0–2 | 3.6 | 0.4 | 3–4 | 3.4 | 0.3 | 3–4 |
| Epidermal mucocytes condition | 1.7 | 0.5 | 1–2 | 4.3 | 0.5 | 3–5 | 4.3 | 0.6 | 3–5 |
| Mesenterial filament mucocytes | 2.7 | 1.1 | 1–5 | 4.4 | 0.7 | 3–5 | 4.2 | 0.9 | 2–5 |
| Degeneration cnidoglandular bands | 1.5 | 1.3 | 0–5 | 4.3 | 1.0 | 2–5 | 3.8 | 1.3 | 2–5 |
| Dissociation mesenterial filaments | 0.5 | 0.9 | 0–3 | 2.8 | 1.5 | 0–5 | 1.9 | 1.2 | 0.2–3.7 |
| Costal tissue loss | 0.3 | 0.5 | 0–1 | 3.5 | 1.3 | 1–5 | 3.2 | 1.4 | 0.9–4.8 |
| Calicodermis condition | 1.4 | 0.6 | 1–3 | 4.0 | 0.7 | 2–5 | 3.8 | 0.9 | 2.1–4.9 |
| Bacterial aggregates | 0.0 | 0.0 | 0–0 | 0.0 | 0.0 | 0–0 | 0.0 | 0.0 | 0–0 |
| Epidermal RLOs | 3.2 | 0.6 | 2–4 | 3.6 | 0.5 | 3–4 | 3.4 | 0.5 | 2.5–4 |
| Filament RLOs | 2.8 | 0.5 | 2–4 | 2.8 | 1.2 | 1–5 | 2.9 | 0.9 | 2–5 |
| **Percent affected (presence/absence)** | | | | | | | | | |
| Coccidia | | 10 | | | 14 | | | 10 | |
| Calicodermis repair | | 0 | | | 43 | | | 33 | |
| Necrotic cell spherules | | 0 | | | 33 | | | 33 | |
| Zooxanthellate ciliates | | 0 | | | 24 | | | 24 | |
| Non-zooxanthellate ciliates | | 0 | | | 10 | | | 14 | |
| Oocytes | | 10 | | | 10 | | | 5 | |
| Spermaries | | 0 | | | 0 | | | 0 | |

margin. The calicodermis varied in thickness and condition, but deeper and closer to the tissue loss margin was thinner, had fewer cells, and calicoblasts lysed or sloughed off the mesoglea (Fig. 6H); sometimes foci of hypertrophied columnar calicoblasts with apical fine acidophilic granules were present at lysing tissue margins. None of these samples had bacterial aggregates, but all had suspect RLOs in mucocytes of the oral disc, tentacle epidermis, and cnidoglandular bands, and infected mucocytes were also present in gastrodermis lining the gastrovascular canals and mesenteries (Figs. 6I and 6J). Suspect RLO cells filling epidermal mucocytes were usually large (1–2 μm) and pleomorphic (Fig. 6J), whereas those in gastrodermal mucocytes were usually smaller (0.2–1 μm) and coccoid (Fig. 6I) and those in cnidoglandular band mucocytes could be either morphology or size within a particular cell.

Tissue loss margins displayed lysing coral cells with vacuolation and necrosis or apoptosis of cells remaining on the skeleton and sloughing of epithelial cells from mesoglea. Some agarose-enrobed samples had free-swimming ciliates containing zooxanthellae on the denuded skeleton in 24% of samples, but were very rarely in contact with coral tissue or within the lumens of gastrovascular cavities or canals even near the tissue loss margin; ciliates without zooxanthellae were present in fewer numbers on 10% of samples, but farther away from tissue remnants. In addition, circumscribed masses of necrotic cell debris and zooxanthellae, in various states of further degradation and lysing, were present in 33% of the diseased samples. About 1–2 mm in diameter, they appeared to form as calicoblasts surrounding gastrovascular canals released from the skeleton and mesoglea surrounded gastrodermal cells or mesenterial filaments or epidermis fragments, trapping the degenerating epithelial cells within, but eventually lysing and breaking apart. All of the diseased samples obtained from colonies used in the mitigation treatments had similar pathological changes (Table 4). Early to mid-stage developing oocytes were found in 10% and 5% of the samples, respectively, but no spermaries were observed.

Evaluation of the frequency distributions of the data to determine normality revealed that most parameters had a bimodal distribution, divided between the apparently healthy and diseased tissues (Table S2), so the distributions were further examined within these categories. For example, Epidermal Mucocytes Condition had no overlap in scores, with apparently healthy samples showing mostly mild changes and diseased mostly severe changes. Parameters with minimal overlap included General Condition $100\times$, Zooxanthellae Condition $100\times$, Dissociation of Mesenterial Filaments, Costal Tissue Loss, and Calicodermis Condition. Parameters with broader frequency distributions of similar scores for both diseased and apparently healthy samples included Mesenterial Filament Mucocytes, Degeneration Cnidoglandular Bands, and Epidermal and Filament RLOs.

Comparison of the apparently healthy samples with all diseased samples (Fig. 7A) revealed that all parameter scores were significantly different, except for Epidermal and Filament RLOs ($p = 0.165$ and 0.767, respectively, $t$-test, Table S3). Epidermal RLOs were judged to be moderate to marked in severity; filament RLOs were mostly judged to be minimal to marked in severity in both groups. For the samples in the mitigation experiment (Fig. 7B), histological parameters were significantly different between

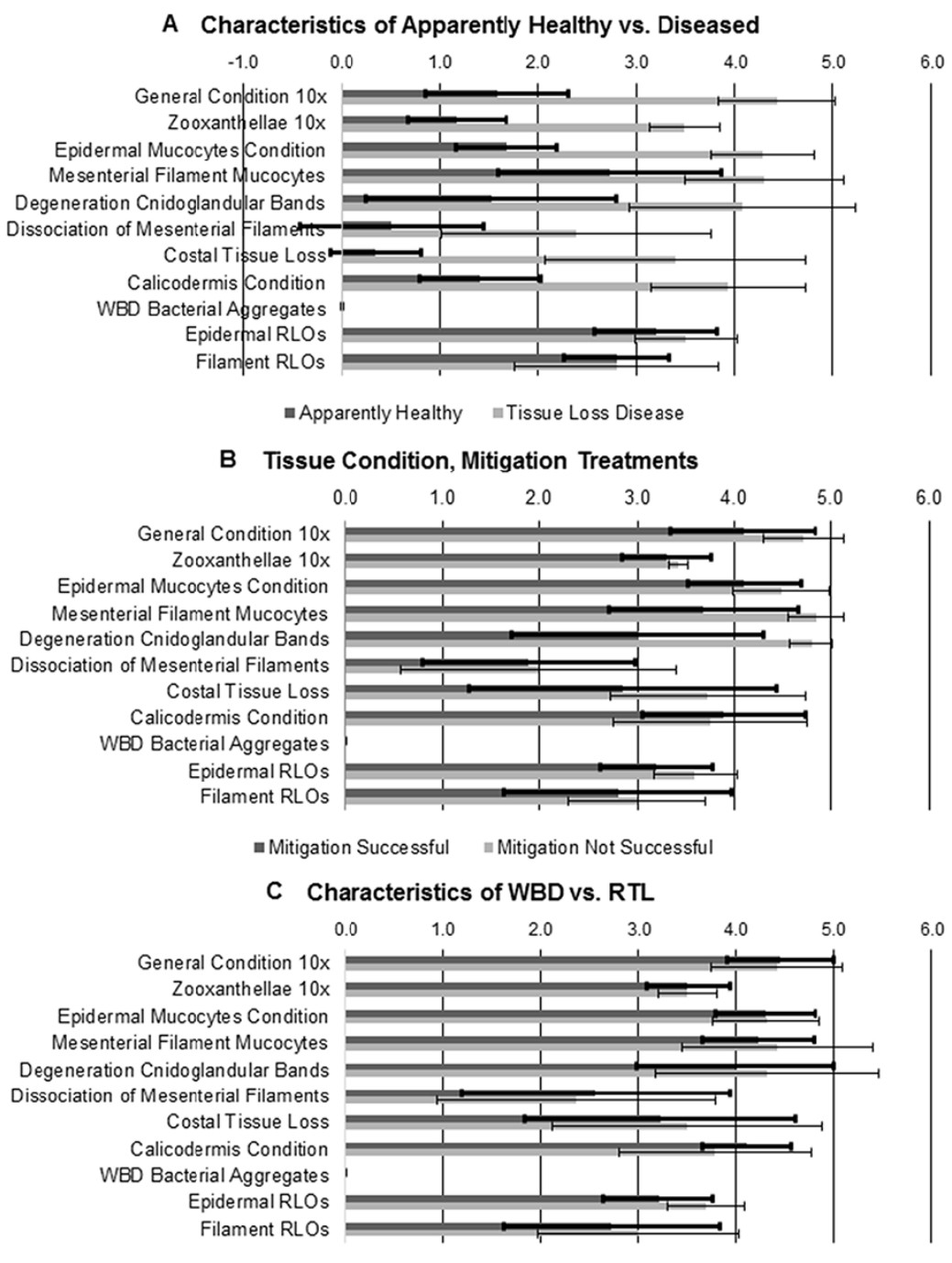

**Figure 7 Histology parameter scores comparisons.** (A) Apparently healthy samples vs. diseased samples. (B) Successful vs. unsuccessful mitigation treatment samples. (C) Microscopic characteristics of WBD vs. RTL samples.

successful and unsuccessful treatments only for Mesenterial Filament Mucocytes and De-generation of Cnidoglandular Bands ($p = 0.0097$ and $0.017$, respectively, Mann–Whitney $U$-test, Table S3). Number of mucocytes in the filaments was markedly fewer in samples from colonies where mitigation was not successful, in addition the filament epithelium had moderate to severe atrophy, loss of cnidocytes and acidophilic granular gland cells, and necrosis or apoptosis of remaining cells. Samples categorized as WBD or RTL in their patterns of tissue loss (Fig. 7C) only differed in Epidermal RLOs scores ($p = 0.031$, Mann–Whitney $U$-test, Table S3).

## DISCUSSION

Surveillance of multiple wild and restored populations of staghorn coral in the Florida Keys during two years emphasizes the severe, ongoing disturbance that disease invokes in this endangered species. Devastating disease outbreaks appear intermittently in both wild and restored patches that have appeared healthy for a number of years. For example, colonies at all three wild sites and restored colonies at Key Largo Dry Rocks appeared healthy with minimal partial mortality that was mostly attributable to fireworm predation throughout the 2011 surveillance. However, two of these four sites (one wild, one restored) were devastated by disease in 2012. All apparently healthy and diseased samples collected in both years were infected with a microorganism that we believe to be, based on morphology and staining with Giemsa, the *Rickettsiales*-like bacterium found by *Casas et al. (2004)* using molecular techniques (Table 4, Fig. 7A). Although *Casas et al. (2004)* dismissed this microorganism as a potential pathogen of staghorn corals because it was present in apparently healthy and diseased samples, as well as other coral species, our histopathological examinations revealed that this microorganism is infecting polyp mucocytes. While a ubiquitous infection in the absence of gross disease signs might be interpreted as commensalism or mutualism rather than parasitism, it could also mean that the infection is still in an early stage or that the coral has been able to maintain its tissues. The intensity of the infection in mucocytes also raises a third possibility that this infection may be altering the coral's mucous secretions and hence, increasing the susceptibility of the coral to other environmental stressors and tissue loss. This third scenario would suggest a ubiquitous compromised health condition affecting the population. There is no evidence that disease dynamics nor histological characterization are different between wild and restored colonies within the study population, which indicates that different disease risk management would not be warranted.

The high rates of disease prevalence documented in the current study (Fig. 3) are not unprecedented as overall average disease prevalences of more than 25% have been reported for individual site surveys in Panama, Belize, Cayman Islands, St. Thomas USVI, Antigua, and Curaçao for *A. cervicornis* (*Fogarty, 2012*; *Vollmer & Kline, 2008*) and for *Acropora* spp. (*Ruiz-Moreno et al., 2012*). However, the peak disease prevalence observed in the current study (∼70%) is substantially higher than reported in these other Caribbean studies. Somewhat lower, but still substantial, average levels of *Acropora* spp. disease prevalence (8–12%) are reported in multi-year, Caribbean-wide, general coral condition surveys

(*Marks & Lang, 2007*; *Ruiz-Moreno et al., 2012*). In comparison, disease prevalence in Pacific acroporid corals are reported in the range of 9–13% (three sites in the Great Barrier Reef, *Willis et al., 2004*), or 2–10% (one site in the Northwest Hawaiian Islands, one site in American Samoa; *Aeby et al., 2011*) while more extensive surveys in three years across the entire Indo-Pacific region indicate an acroporid disease prevalence of around 4% (*Ruiz-Moreno et al., 2012*).

The existence of disease-resistant genets within *A. cervicornis* has been reported at a frequency of 6% in a studied population of 49 genets in Panama (*Vollmer & Kline, 2008*). Four of the restored populations surveyed in this study are in fact genotypically depauperate, containing the same three genets, while the other two restored populations were genotypically more diverse (Table 2). Colonies at the three wild sites have not been genotyped, but multiple genets and high genetic diversity have been previously documented in wild populations of *A. cervicornis* in the Florida Keys (*Baums et al., 2010*; *Hemond & Vollmer, 2010*). Thus, it is likely that multiple genets were present in each of these sites as well. The detection of potentially disease-resistant genets seems problematic in these populations. Among the three wild sites, we might have surmised possible disease-resistant genets within these presumably genotypically-diverse patches given low disease prevalence in 2011. However, all of the tagged colonies at wild site Little Conch were observed with disease at some point during 2012 (Table 3). An important goal of Caribbean *Acropora* population enhancement strategies is the nursery culture of stress-resistant genotypes or phenotypes in order to propagate hardier restored populations (e.g., *Bowden-Kerby & Carne, 2012*). The current results showing (1) extreme variation in disease manifestation over sites and years, and (2) generally lower manifestation of disease within the nursery environment than in nearby reef outplanted populations (despite similar suspect RLO infection levels), reveal a challenge in accurately identifying these hardier candidates while emphasizing that the environmental factors limiting survivorship of *A.cervicornis* persist in the 'wild' reef habitat of this region.

Similarly, the site-specific nature of both disease prevalence and incidence patterns (i.e., patchy but not spatially autocorrelated) challenges the hope of identifying specific environmental triggers for disease, at least on the site scale. While no severe warm thermal anomalies occurred during the duration of this study, accumulated thermal stress (e.g., duration of exposure >30 °C) was greater in 2011 than 2012 (Figs. 3E and 3F)—corresponding to mild bleaching observed in some wild colonies during September–October 2011 (none in 2012)—but not greater disease impacts. Previous and repeated reports of *A. cervicornis* disease in the Florida Keys have occurred in late spring to mid-summer (April–July; *Williams & Miller, 2005*; K Nedimyer, pers. comm., 2004; M Miller, pers. obs., 2009), not coinciding with the seasonal temperature peaks which occur in September–October. The only coherent spike in disease prevalence and incidence that was discernible across all sites corresponded to the passage of Tropical Storm Isaac (Fig. 3), corroborating the hypothesis that storm disturbance may be an important coral disease trigger (*Brandt et al., 2013*; *Bruckner & Bruckner, 1997*; *Knowlton et al., 1981*; *Miller & Williams, 2006*).

The only significant difference we were able to discern between restored and wild colonies was in the degree of partial mortality during the storm interval, with restored colonies having greater partial mortality than wild colonies (Fig. 4). One limitation of the current study is in the spatial confounding of the restored and wild sites, with the former restricted to more exposed, mostly shallow fore-reef habitats and the latter in somewhat more sheltered patch reef habitats. It is likely that this habitat difference accounts for the apparent greater vulnerability of restored colonies to storm-associated disease mortality rather than any inherent characteristic of the colonies.

Our mitigation tests did not detect any significant benefit, in terms of preventing tissue loss over a four-week period, from either excision or epoxy-band treatment. However, high variability in response of both treatments, as well as the controls, yielded low power in the statistical tests and several other observations may affect the interpretation of the somewhat inconclusive results. First, there was no hint of harm accruing to either treatment (Fig. 5). Secondly, during circumstances of high disease prevalence we commonly observed, a 'successful' excision (as observed one month after treatment) or another area on a successfully epoxy-banded colony to resume tissue loss at a later time. This suggests re-activation of disease (or reinfection with a pathogenic microorganism) can occur in a given colony in environments with high disease 'load'. On the other hand, if treatment replicates that were implemented at times and sites with high prevalence (arbitrarily set at $>15\%$) are excluded, the remaining replicates indicate significantly lower frequency of tissue loss for treatments (especially excisions) vs. controls ($X^2$ test; $p = 0.014$; see Table S4). Our results and observations suggest that if mitigation interventions are attempted, branch-tip excisions are perhaps more likely than epoxy bands to be successful. Histologically, tip tissue may be in better condition than that at the tissue-loss margin and resources are directed toward the tips rather than bases in this species (*Highsmith, 1982*). Also, mitigation appears to be more successful in isolated cases rather than in areas with more disease. Unfortunately, conditions with low disease prevalence (arbitrarily examined as $<15\%$) occurred in only 31 of our 56 individual site surveys in 2012.

The histopathological examinations revealed several other explanations for variation in mitigation treatment success, despite the challenges in assigning a semi-quantitative score to observations constituting a continuum. The only significant differences in scores between the successful versus the unsuccessfully treated branches were the greater loss of mesenterial filament mucocytes and degeneration of the cnidoglandular bands of the filaments in samples from colonies that had unsuccessful treatments. The filament epithelium lines the free edges of mesenteries in the gastrovascular cavity below the actinopharynx in the polyp; the specialized acidophilic granular gland cells of this epithelium release enzymes to break down food particles. The number and size of gland cells and mucocytes in the cnidoglandular band increase, whereas ciliated cells decrease, aborally in normal *A.cervicornis* tissue. Cell loss, necrosis, and lysing in the cnidoglandular bands indicate that the polyp is no longer able to process particulate food in the gastrovascular cavity. In addition, although condition of the zooxanthellae, which also supply nutrients to the coral, appears to remain unaffected until the host

tissue is sloughing off the skeleton, their numbers are reduced as the host condition deteriorates. However, due to our inability to visually detect changes in coral pigmentation until zooxanthellae numbers are reduced by more than 50 percent (e.g., *Jones, 1997*), the tissue grossly appears to be intact and "normal", when it may not be so microscopically. The ubiquitous presence of the suspect RLO infections and their apparent association with mucocyte stress and loss raises the possibility that most, if not all, the *A. cervicornis* population's health is compromised. Thus, without microscopic examination, it is difficult, if not impossible, to identify the "best candidates" for mitigation treatment.

Exactly what the impact of the suspect RLOs is on the *A. cervicornis* colonies is conjecture at this point, but based on the behavior of similar obligate intracellular bacteria, their replication within host cells requires substantial energy (*Fryer & Lannan, 1994*) resulting in tissue atrophy and necrosis (*Friedman et al., 2000*; *Sun & Wu, 2004*). Nutritional stress may be a primary reason why the zooxanthellae are gradually lost and calicoblasts lyse (*Schoepf et al., 2013*; *Weis, 2008*). The coral cannot maintain its tissues with the loss of these host and algal cells that are crucial to its survival. Infected mucocytes eventually die and are released from the epithelium, as seen in this study, and the coral may not be able to replace them if nutritional resources are compromised. Reduction in mucocytes means the loss of the coral's protection against sedimentation and microorganisms, as well as heterotrophic feeding (*Brown & Bythell, 2005*; *Ritchie, 2006*). Investigation of the pathogenesis of the suspect RLO infection is continuing, noting that other bacteria (*Vibrio harveyi*, *Serratia marcescens*, unspecified) have been implicated in the acute loss of tissue from Caribbean acroporids (*Gil-Agudelo, Smith & Weil, 2006*; *Kline & Vollmer, 2011*; *Patterson et al., 2002*). Transcriptome analysis shows gene expression alterations in immunity, apoptosis, cell growth, and remodeling in WBD (*Libro, Kaluziak & Vollmer, 2013*); and multiple pathogens may be involved or be different in specific cases requiring histopathological examinations (*Work & Aeby, 2011*). However, ciliates do not seem to be a major factor in tissue loss in our study. Bacterial aggregates first proposed to be the pathogen (*Peters, Oprandy & Yevich, 1983*) were not present in any of these samples. *Work & Aeby (2014)* observed diverse cell-associated microbial aggregates (CAMA) in Indo-Pacific corals and concluded that they were benign or beneficial to the hosts; however, this may be premature, since long-term studies have not been undertaken in most of the coral species. *Anderson et al. (2003)* reported that the formation of intracellular biofilm-like "pods" of *Escherichia coli* within the epithelium lining the urinary bladders of mice had a role in chronic bladder infections. Histologically, no differences could be discerned between WBD- and RTL-affected colonies, suggesting that differences in the patterns of tissue loss (Table 1) are due to the intensity and duration of suspect RLO infections or the identity of other biotic or abiotic stressors that trigger the loss. Thus, field identification of diseased *A. cervicornis* lesions should be limited to acute or subacute tissue loss and the patterns of distribution (e.g., focal, multifocal, diffuse). Samples collected from the same colonies in this study are also being processed for molecular characterization of the microbial communities associated with them at the diseased margin

and in apparently healthy tissue from diseased or unaffected colonies, as well as performing transmission electron microscopy, to help explain the pathogenesis of tissue loss.

Overall, our results confirm the devastating toll that disease continues to have on both wild and restored populations of Caribbean staghorn coral and suggest that wild and restored populations display similar disease conditions, dynamics, and impacts. These results emphasize the continuing need to understand and effectively address disease impacts in this species, as well as discover methods and run experiments to try and determine a way to minimize tissue loss of diseased colonies. Unfortunately, the straightforward mitigation treatments tested in this study provided ambiguous results. Given these results, population restoration might be viewed as a necessary but stop-gap recovery measure, particularly in light of the suspect RLO infections of mucocytes in nursery and wild colonies. Additional assessments of factors affecting the staghorn corals and their tissue loss diseases are needed, including pathogen interactions between the stocks (*Friedman & Finley, 2003*) and host genotype susceptibility (*Vollmer & Kline, 2008*).

## ACKNOWLEDGEMENTS

This study was made possible by support from UNCW/Aquarius Reef Base program. The work would not have been possible without the invaluable collaboration and support of K Nedimyer (Coral Restoration Foundation) for which we are truly honored and grateful. Additional assistance in the field from O Rutten, T Roberts, A Bright, and C Kiel is greatly appreciated. S Kang, P Pansoy, and W Norfolk provided support in the George Mason University Histology Laboratory.

### Funding

This work was funded by the NOAA Coral Reef Conservation Program via the Aquarius Reef Base Program. The funders had no role in study design, data collection and analysis, decision to publish, or preparation of the manuscript.

### Grant Disclosures

The following grant information was disclosed by the authors:
NOAA Coral Reef Conservation Program.

### Competing Interests

The authors declare there are no competing interests.

### Author Contributions

- Margaret W. Miller and Esther C. Peters conceived and designed the experiments, performed the experiments, analyzed the data, contributed reagents/materials/analysis tools, wrote the paper, prepared figures and/or tables, reviewed drafts of the paper.
- Kathryn E. Lohr conceived and designed the experiments, performed the experiments, analyzed the data, wrote the paper, reviewed drafts of the paper.

- Caitlin M. Cameron conceived and designed the experiments, performed the experiments, prepared figures and/or tables, reviewed drafts of the paper.
- Dana E. Williams performed the experiments, reviewed drafts of the paper.

## Field Study Permissions

The following information was supplied relating to field study approvals (i.e., approving body and any reference numbers):

Study was conducted under Florida Keys National Marine Sanctuary Permit #FKNMS-2011-032-A1.

## Supplemental Information

Supplemental information for this article can be found online at http://dx.doi.org/10.7717/peerj.541#supplemental-information.

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
