# Peer review of "Disease dynamics and potential mitigation among restored and wild staghorn coral, Acropora cervicornis"

_PeerJ, doi:10.7717/peerj.541_

## Round 0.1 · original submission · Minor Revisions

· Academic Editor

Minor Revisions

I am sorry for the delay in getting this manuscript back to you, but the first two reviews came back with essentially opposite recommendations, so I asked for a third. As you can see, the third review falls between the first two, but with a similarly positive take on the suitability of the manuscript for submission as the second and a similar concern for clarification of the manuscript as the first. I am inclined to agree with the two more positive referees that the manuscript deserves to be published, but hope that the authors will take the comments of the more critical referees to heart in revising the text. In particular, the first reviewer has a number of good points and highlights several areas throughout the text where revision for clarity could prevent future readers from having a similar reaction. The third referee has included a marked up text with suggestions on the original PDF for your consideration in revision.

Overall, I would like to see your response to the referees and will decide whether I need to return it to them for final feedback when I see the revisions. I believe that the comments can be addressed with careful revision, and I look forward to seeing your revised manuscript.

·

Basic reporting

See general comments to authors.

Experimental design

See general comments to authors.

Validity of the findings

See general comments to authors.

Additional comments

Disease dynamics and potential mitigation among restored and wild staghorn coral, Acropora cervicornis. by Miller MW, Lohr KE, Cameron CM, Williams DE, Peters EC.

Overview: The authors set out to compare wild versus reintroduced populations of caribbean Acropora by characterizing tissue loss disease over time and at the cellular level and to evaluate various methods of intervention to prevent progression of lesions. There is probably valuable information struggling to get out of this paper, but the presentation was so convoluted and the experimental designs so opaque that it was really hard for the reader to get a sense of what points the authors were trying to convey. Indeed, the paper almost reads as a "data dump" with not very much thought placed on carefully synthesizing and summarizing key points. The authors really need to step back and completely rethink and reconsider exactly what is their bottom line message and why this study is important. Specifics follow:

Line 46: One could argue that it is not so much the lack of diagnostic tools for corals (although that is an issue) but rather their improper implementation that has led to a vaccuum in knowledge on potential etiologies of coral disease in the Caribbean. For example, your study is unusual in that it rightly incorporates histology to assess what is going on.

Line 78-79: You are either outplanting corals from nurseries or transplanting from one location to another in the wild. In order to assess risk of those activities to wild populations, I would imagine there must be some sort of health monitoring going on in both coral nurseries and source wild populations to gauge the risk of translocating infectious agents. If so, it would be informative for reader to learn what that is. If not, then how is one to gauge success of translocations? For example, if translocations fail, how do you know it was not something lurking in the rearing facility that bloomed once corals underwent the stress of explanting?

Line 81: OK, how similar are the restored vs wild sites in terms of substrate, depth and location on reef. I see here restored sites are on forereef but not clear about wild site...is that on reef slope? Crest? Is substrate rubble for both? Seems that this may be important to judging whether comparisons are fair or not.

Lines 95-133: There is a long diversion here on discussing various kinds of WBD, it is evident here that authors are really not sure what is what, and it is really not lending clarity to the story. For example, one of your criteria for defining WBD depends on histopath (Gladefelter) so how do you make the call of what lesion type is is in the field? Line 123: Precisely-given that these lesions change over time, they could look like RTL early on or WBD later. Seems all very artificial and arbitrary. How about simplifying, eliminating all this verbiage, and distilling down to three categories: Acute tissue loss, subacute tissue loss (where bare white skeleton progresses to algae cover) and predation? That way, you are not tying yourself up in knots over vague and possibly contradictory definitions, and you are basically dealing with only 3 types of lesions (those unexplained by predation-subacute or acute tissue loss) and those probably explained by predators (snails, worms). You kind of go on to do this anyway further down the MS (line 147), so why not be explicit about it?

Line 135: Not sure how the Willis citation (for Pacific corals) really applies to seasonality of disease in Caribbean Acropora.

Line 140: It seems from earlier statements that your survey methods really depended more on getting a minimum number of colonies rather than a specific survey area, so why not simply do away with radius plots?

Line 142: Recommend you stick to chacterizing lesions and not "disease" per se, because that is really what you are doing.

Line 148: Wait, do you mean that you inferred predation in some cases of tissue loss when predators were not around? How then to sort this out from unexplained tissue loss? Typically, predation is inferred when one sees the culprit nearby.

Line 150: If not counted as disease, then lesions associated with predators were called "predation" no?

Line 152: I'm not really clear here on analysis. I understood the wild sites were going to be reference for the outplanted sites? What makes nursery site an appropriate reference?

Line 160: How was this percent tissue loss estimated? guidance? For example, did you have estimated bins..0-30%, 30-60%, etc.?

Line 203: Why were cases eliminated if corallivores came on the colony? Isn't predation one of your metrics? Might it not be valuable to calculate incidence of predation as well as unexplained tissue loss to help explain its relative importance?

Line 211: I'm confused here. What is the mitigation experiment and how is that different than survey sites? Were disease samples collected from outplant sites?

Line 212: Any reason why paired healthy samples were not collected concomitantly with disease samples? That might have made for a fairer comparison.

Line 232: What other staining procedures was used?

Line 235: What do you mean by condition? Are you referring to post-mortem condition? Or number of zoox present? Or general intactness of tissues? Pls clarify.

Line 237: "Within normal limits"...unfortunately, this definition might vary with investigators. It becomes really difficult here to interpret the histology table. For example, if I were to look at a slide for zoox, how would I differentiate mild from moderate from marked from severe depletion?

Line 242: RLOs....Casas et al. ID'd these from molecular and did not do microscopy. So they really did not actually visualize the organisms associated with host cells. Given that, what were your criteria to define something in the tissues as RLOs?

LIne 244: VLPs...given that convincing evidence of virus-induced diseases in corals is still being awaited, I would be keen to see photos of VLP in tissues. Depending on personal communications seems a bit shaky.

Line 246: How were gonads staged? Can you pls provide citations or criteria?

Line 250: There are other stains above and beyond Giemsa that would be far more useful to identify possibility of RLO. Machiavello or Giminez stains both are pretty specific for rickettsia and would be more appropriate than Giemsa. And if you really think you have RLOs, then electron microscopy would be useful to confirm morphology.

Lines 258-270: Consider collapsing all wild and all restored sites into a single category and comparing trends between the two.

Line 271-280: I'm really confused here as to what you are trying to tell me. What is accumulated thermal stress? How did the storm relate to temperature?

Line 281-288: OK, so there is variability in prevalence of lesions between sites....this is self evident and is somewhat akin to saying the sky is blue. So what? Why is this important?

Line 288-A total of 89 new cases?

Line 293: What does very similar mean exactly?

Lines 307-375: The whole histology results really need to be rethought. COnsider condensing to the bottom line points you want to communicate and not drown the reader in histological detail. See my comments on Table 4 and Figure 6.

Line 338-How did you differentiate apoptosis from necrosis? What was the size range of RLOs? What percent had ciliates in contact with intact tissues? Were these invasive? Sounds like ciliates were really secondary players in lesion. LInes 345-347 seems somewhat speculative....how do you know permeability to seawater was a factor?

Line 386: This is really speculative. First, I'm not entirely convinced that RLOs are really there and second, making the leap that they are affecting mucus production is really stretching it.

Line 400: May also want to include following reference: Aeby GS, Bourne DG, Wilson B, Work TM (2011) Coral Diversity and the Severity of Disease Outbreaks: A Cross-Regional Comparison of Acropora White Syndrome in a Species-Rich Region (American Samoa) with a Species-Poor Region (Northwestern Hawaiian Islands). Journal of Marine Biology 2011:490198. They found 2-10% Acropora affected with dz.

Line 401-419: Consider deleting this....you really did not design your study to evaluate genet effects on dz.

Line 420: What is accumulated thermal stress? Seems like lines 420-434 could be condensed into a couple of sentences. One saying that temperature had no evident effect and another highlighting other work on temperature vs tissue loss dz in Acropora.

Lines 435-442: Indeed, really hard to disentangle effects of storm versus location. How do we know that storm effect was not greater due to location of colonies.

Line 443-464: So after all this, what is the real bottom line for intervention to stop disease. Suggest all this be condense do the salient 1-2 sentences which is, after all this effort, hard to conclude.

Line 473: I doubt in prep reference will be accepted in PeerJ but maybe I'm wrong.I think all this talk about nutrition is really speculative based on data. You really saw no significant changes between groups for intervention, and that's OK, but no need to go through a lot of hand waving.

Lines 485-508: Given the uncertainty about whether or not the RLOs are really even prokaryotes, suggest delete.

Lines 509-519: I would argue that given the low difference between reintroduced and wild, there is a crying need to redouble efforts to sort out what is going on at cellular level to explain mortality of Acropora in Caribbean.

Table 1. Lots of NAs for treatment group sites. Does it really make sense to examine treatment effect by site given low sample sizes? Suggest you completely collapse all sites to look at treatment effects and delete Table 1 altogether. Who careas about genets in this particular study?

Table 2: Delete....you don't really analyze data by lesion type other than explained (predation) and unexplained tissue loss, so why add all this superfluous material?

Table 3: What intervals merit a particular shading scheme? I'm also not clear here in Little Conch. If unaffected is 0, that means that 100% of colonies were affected, yet incidence is rarely above 10% for each time period....how can that be?

Table 5. Also, would be useful to repeat your categories (0-5) and what they mean in the table heading. What is general condition? Does epidermal mucocyte condition refer to relative fill of mucocytes with mucus? Number of mucocytes? What is the cellular change that signifies tissue loss? Absence of tissues? What is calicodermis condition? Are you referring to degree of hypertrophy? Are ciliates actually invading intact tissues or are they hanging out in associated debris?

Figure 1 b. I would say those lesions are more diffuse than just multifocal.
Fig. 1c-d. Other than distribution (localized vs multifocal), what distinguishes WBDI vs RTL? In Fig 1D, one could easily argue that this is early stage WBD. Fig 1E. Again, seems artificial to have RTL and WBD as could be different stages of same process. Fig. 1F Hard to appreciate bleaching of tissues in 1F...looks more like subacute tissue loss.
In all honesty, if I were doing field surveys, I would have a real hard time justifying the assigning of different names to these lesions that are basically all variations of acute tissue loss with different patterns of distribution on the colony (diffuse, locally extensive, mutlifocal, or combination). The way you have things defined above in your figure (other than predation) does not really lend itself to consistent replication among investigators.

Figure 3. For clarity of presentation, might it not make more sense here to simply collapse wild site and restored sites into two separate categories and lot prevalence over time? Seems that would make more sense anyway since isn't that your basic question (do corals restored fare better than wild?)

Figure 4. Rather than partial mortality, suggest use percent tissue loss. There is no such thing as partial mortality.....mortality is binary....you're either alive or you're dead. Also, it is difficult to compare here because we are looking at actual frequency rather than percentages. Should show percentage of colonies with different percentages of tissue loss no?

Figure 5. Delete....this is really not adding to story significantly.

Fig 6. A) Photos is a bit overexposed making it difficult to appreciate details. Did you stains sections with alcian blue/PAS to rule out possibility that RLOs could be mucocytes? PAS and alcian blue stain sulfated mucopolysaccharides that are common components of mucus. C) Overexposed....pls increase contrast. Inset high mag showing calicoblast detail would help reader better appreciate changes. F) A bit more contrast here would help H) Overexposed. More contrast pls. Consider higher mag shots for C & H to show more cellular details. I-J) Not convinced that these are necessarily Rickettsiae or even prokaryotes. Might be helpful to have oil immersion shot here to really see details on light microscopy.

Figure 7. THis is really hard to interpret. What do the different shadings in the bars mean? Looks to me that there is very little difference between WBD and RTL histologically (that would make sense since I can't possibly see how these two conditions can realistically be differentiated in field settings anyway). Seems the same applies for comparisons of treatments (again, not clear what is what in graph). So that really leaves disease vs healthy where there appear to be some differences. Suggest keep that graph and delete remainder for clarity.

Reviewer 2 ·

Basic reporting

A nicely done study comparing the prevalence and incidence of tissue loss disease in acroporids from nurseries, wild and extant populations. Their methods of study were appropriate and interpretation of the results sound. They tried two methods of disease treatment, both unsuccessful, but important to report so progress can be made in this area (disease treatment). However, they fail to mention two successful methods of coral disease treatment (Dalton et al. 2010, Hudson 2000). These successful cases should be mentioned and an explanation of why they worked yet their attempts did not.

Experimental design

Appropriate

Validity of the findings

The detailed histology was very informative but the conclusions concerning the RLOs that were discovered are highly speculative. Work and Aeby (2014) found that numerous coral genera, including acroporids, have cell-associated bacterial clusters but concluded that they could be of benefit to the coral host. Their conjecture that RLOs may be negatively affecting coral health and could be underlying the problems with tissue loss diseases is not supported by any evidence. That component of the discussion should be removed or minimized. Their statement that disease prevalence was “low” (line 260) with means of 1.5-4.4% was an interesting interpretation as I would consider that a “high” level of disease. I suspect that in the Caribbean there is a tendency for a shifting baseline of what is considered “normal” or “low” disease levels and that should be discouraged. It also needs to be made clear that disease levels in other coral genera are much lower and that is what should be considered “low” or “normal”. Maybe qualify their statements as saying “for acroporids, disease levels were low” or something along those lines.

·

Basic reporting

No Comments

Experimental design

The structure and implementation of the experiment is sound and justified. However, some clarifications would be helpful. For example, in the ‘restored’ sites why are there corals from the nursery in addition to transplanted wild corals? What proportion of transplants vs nursery corals are in the ‘restored’ sites? How may this impact your conclusions?
There are a few other areas that could use further explanations or clarifications. Please refer to the attached document with edits and comments.

Validity of the findings

The results of this study appear valid and will be a significant contribution to the current state of knowledge regarding disease risk associated with outplanting corals for restoration. Understanding baseline levels of disease within wild populations, disease prevalence within nurseries, and how outplanting activities may or may not influence disease dynamics is critical for conducting informed and successful restoration activities. I do, however, recommend the conclusion stating that RLOs reflect compromised health be a bit tempered. Since RLOs are seen in all samples (healthy and diseased), even apparently healthy samples from the 1970s, they may not actually cause any measurable physiological change on the coral. I believe further investigation is required before you make these conclusions.

Additional comments

The research within this manuscript has been conducted well, has produced significant results, and will be a great contribution to restoration science. I recommend publication after minor revisions. These are all included in the edited document provided.

---

## Round 0.2 · Minor Revisions

· Academic Editor

Minor Revisions

Both referees have now had the opportunity to review your revised manuscript and both are satisfied with the revisions to the point that the manuscript should be acceptable for publication following some minor revisions that are largely editorial in nature. The one remaining point of contention remains whether or not the Giemsa-positive structures play a pathogenic role in tissue loss in Acropora. While I agree with the referee that this seems highly debatable, I also feel that the discussion is the place for authors to express their opinion and to allow some speculation about the likely importance of their work. Having said that, if the authors wish to speculate, I suppose I would urge authors to make clear what exactly is speculation until the concrete evidence that would be convincing to the first referee is available.

Regardless, I feel that I am in agreement with the reviewers that this manuscript is a valuable contribution to the literature, and I do not expect this to go back out to review. I ask the authors to clear up the last few issues suggested by the referees and return the manuscript to me when you are able so that I can make the final decision. I don't expect that it should take long to address these final comments.

·

Basic reporting

See comments to authors

Experimental design

See comments to authors

Validity of the findings

See comments to authors

Additional comments

Disease dynamics and potential mitigation among restored and wild staghorn coral, Acropora cervicornis. by Miller MW, Lohr KE, Cameron CM, Williams DE, and Peters EC

Overview: This is a re-review of the MS. I thank the authors for taking the time to address my comments on previous version. Although I'm not in total agreement with all the rebuttals of the authors, the fact that this MS probably has one of the more comprehensive descriptions of tissue loss in A. cervicornis at the tissue level would lend it merit. The MS has been improved in the sense that it is more clear to interpret. I'll give benefit of doubt here and accept that the Giemsa-positive structures could be RLO (although EM would still be desirable to confirm). However, whether or not these play a pathogenic role in tissue loss in Acropora is highly debatable, and I would urge authors to tone down speculation to that effect until more concrete evidence is forthcoming.

Line 393-There really was no difference between disease and healthy for RLO, so suggesting that these somehow play a role in disease pathogenesis by altering mucus production seems a stretch. Suggest tone down or delete.

Line 398-Might want to add somwhere in there that peak prevalence of disease in your sites was about twice that of highest level reported in literature.

Line 426- Higher disease in transplanted corals in wild also suggest that factors limiting survivorship of Acropora in wild continue to occur in FL.

Line 483-Maybe the RLO are not pathogens but actual symbionts. Were they also seen in samples of Acropora collecetd during 1980s by ECP?

Table S2: "Frequency distributions of condition or severity/intensity scores for tissue parameters of apparently healthy and diseased A. cervicornis samples. Shaded cells show numbers of samples in each category sharing scores." I'm unclear heare what the shading means in the cells. What does this mean by sharing scores? Do you mean that the shaded areas where those where significant differences were not found? Perhaps include an explanatory note such as "For example, clear distinctions were seen statistically between healthy and disease for all condition scores for epidermal mucocytes whereas for mesenterial filament mucocytes, distinctions were less apparent between diseased and healthy as evidenced by more shaded areas)."

·

Basic reporting

Overall, I am content with the revisions provided by the authors for this second round of reviews. I have a few comments, mostly within the discussion, that should be addressed before publication. Most of the suggestions and edits within the revised manuscript are editorial in nature. Please see the uploaded pdf for comments.

Experimental design

See attached document

Validity of the findings

See attached document

---

## Round 0.3 · accepted · Accept

· Academic Editor

Accept

Thanks for taking the time to clarify those last few points in your manuscript as requested by the referees. I feel that you have done a good job of making your points clear and providing context for your conclusions regarding this work. I am happy to accept your manuscript, and move it forward in the publication process.